# Navigating Hidden Hunger: An Exploratory Analysis of the Lived Experience of Food Insecurity among College Students

**DOI:** 10.3390/ijerph191912952

**Published:** 2022-10-10

**Authors:** Ashlyn Anderson, Jacqueline Lazarus, Elizabeth Anderson Steeves

**Affiliations:** Department of Nutrition, University of Tennessee, Knoxville, TN 37996, USA

**Keywords:** food insecurity, college students, lived experience, higher education

## Abstract

College students are a vulnerable population to food insecurity (FI), which has significant implications for academic and health outcomes. The aims of this study were to explore the meaning of FI and its impact on students’ lived experiences and food decisions, facilitators and barriers to food access as a student, and students’ proposed solutions to address FI. Semi-structured, qualitative interviews were conducted with thirty students from a large, public land grant university in the Southeast United States. Grounded theory methodology was utilized with a constant comparative coding strategy to guide thematic analysis. Nine main themes emerged. Themes included the perceived meaning of FI, students’ lived experience with FI, and food related coping strategies and decisions. Facilitators to food access were found to be social-networks and on-campus resources, while barriers to food access included financial burden of higher education, and stigma and social comparison. Proposed solutions to FI aligned with two main themes: food access solutions and information access solutions. Both of these themes included multiple subthemes that provided specific suggestions to address food insecurity for students. The findings aid in understanding the complex lived experience of FI and can inform future efforts to center student experiences, perceptions, and feedback into institutional frameworks to best meet student needs.

## 1. Introduction

Food insecurity (FI) is defined by the USDA as the lack of consistent and reliable access to sufficient, affordable, and nutritional foods to sustain a healthy lifestyle [1]. In 2020, 10.5% of households in the US were food insecure at least some time during the year [1]. A systematic review of FI among students attending higher education institutions in the United States found FI rates to be an average 43.5% [2]. Another scoping review estimated FI among college students to range from 10–75% [3]. While variation in assessment protocols and survey metrics remain [4], FI prevalence among college students is well documented and is significantly higher than the household average. The prevalence and correlates of FI among college students in higher education institutions across the United States clearly indicate that this should be a public health priority [5,6,7,8].

One potential reason that we are seeing problematically high rates of FI among college students may be that the demographics of college students have diversified over time. Over 70% of college students are now considered non-traditional, including students who are first-generation college attendees, have extended time since high school graduation, or come from lower socioeconomic backgrounds [9]. These students may be at higher risk of experiencing FI, as we know that FI disproportionately impacts some college students such as first generation college students [10]. Additionally, students from racial and sexual (LGBTQ+) minority groups are at higher risks of experiencing FI, exposing disparities in FI rates across population groups [11].

A growing body of literature has established college students as a population vulnerable to the detrimental effects of FI. Food insecure students were found to have unhealthy coping strategies including consuming low quality diets, purchasing inexpensive, processed food, and stretching food to make supplies last [12,13]. Basic needs insecurities such as not having adequate and/or stable housing are also closely linked with FI and have been shown to affect college student mental health, academic outcomes, and graduation rates [14,15]. Other physical and mental health effects of food and basic needs insecurities include poorer self-reported physical health, symptoms of depression, and higher perceived stress [16,17]. Further, graduate students have been found to have distinct experiences with FI in both prevalence and characteristics including heightened depression, anxiety, and stress [18,19]. 

A handful of food insecurity interventions have been implemented across higher education institutions with limited degrees of evidence-based effectiveness [5,20]. Federal food resources such as the Supplemental Nutrition Assistance Program (SNAP) have been shown to decrease FI, yet participation in the program by college students remains low due to eligibility requirements, application burden, and stigma [9,21]. Though other campus food resources such as food pantries are prevalent, lack of awareness and barriers to access persist. A critical need for student-centered intervention strategies remains [22,23].

Despite the proliferation of studies on FI prevalence in higher education [2,5,6], an in-depth understanding of the meaning and lived experience of FI among college students remains lesser known. To date, a relatively small body of qualitative research has been published on the lived experience of college FI and student suggestions for addressing the issue in higher education [24,25,26,27,28]. Gathering student stories and lived experiences through qualitative methods is necessary for future research to center students experiencing FI by leveraging their voice as a key informant for change and interventions [27]. To address the gaps in the literature, this research sought to explore the meaning of FI for students in the way it impacts lived experiences and food decisions, perceived facilitators and barriers to food access as a college student, and proposed solutions on how to address FI. 

## 2. Materials and Methods

### 2.1. Study Population 

This study was conducted at a large, public land grant university in the Southeast United States. Participants were recruited from a database created from a previous study on FI among college students [29]. Students who screened as food insecure in the previous study were randomized and sent recruitment emails. Students were contacted twice (one initial contact and one follow up). Interested students completed an electronic screening survey to assess eligibility. Participants were determined eligible if they met the following criteria: (1) were 18 years of age or older, (2) spoke conversational English, (3) enrolled as a student at the participating University, and (4) screened as food insecure on the 6-item USDA food security assessment [30]. All students scored as having low or very low food security (scores ranged from 2–6). The sample included sophomore through senior undergraduate students and graduate students. Freshman students were excluded because at the time of study initiation (fall semester 2020) these students would have been new to campus and had limited lived experience as college students. All study procedures were approved by the University’s IRB (UTK IRB-20-05905-XP). Informed consent was obtained from all students, and all interviews were conducted by phone or on Zoom. Participants received a USD 25 gift card for their participation in the study.

### 2.2. Data Collection 

A qualitative, semi-structured interview guide was developed through an iterative process based on the research aims and a literature review of the existing qualitative research on food insecurity among college students. The interview guide was reviewed by the co-authors, additional research team members, and a panel of five PhD-level experts who are knowledgeable about food security, nutrition, college student populations, and qualitative research methods. The guide was iteratively discussed and revised by the co-authors after each round of review. The finalized interview guide topics were selected because they aligned with the research aims to identify student meaning and lived experiences with FI, perceived facilitators and barriers to food access, and proposed solutions to address FI. Questions were developed to assess emic terminology of students around food insecurity, and beliefs surrounding their food situations to better understand areas of sensitivity and stigma that impact resource utilization. Selected interview guide questions can be found in Appendix A. 

The interview guide was pilot tested with 3 college students attending this University, but not eligible for this current study, prior to starting data collection. No major issues with the guide were identified in the pilot testing. Minor revisions to enhance use of plain language were identified during the pilot tests and made, and the guide was considered ready for use. Semi-structured interviews were conducted by a trained researcher from October 2020 to April 2021 (fall and spring semesters). Interviews lasted approximately 45–60 min (mean duration 54.83 ± 8.44 min). Demographic characteristics were collected at the end of each interview including participant age, race, gender identity, first generation student status, meal plan usage, and housing situation (Table 1). Data saturation, in which emergent themes were repetitive to prior interviews, was reached at 28 students; 2 more interviews were conducted to validate saturation for a total of 30 interviews [31]. Interviews were audio recorded and transcribed verbatim using the Zoom software transcription feature, and transcripts were cleaned by the interviewer and an independent research assistant to check for errors, then uploaded into NVivo (QSR International, Melbourne, Australia, version 1.6.1) for analysis. 

### 2.3. Data Analysis

This study was conducted using a grounded theory methodology, in which the data of participant voices drove the creation of a codebook as an iterative process [32]. Transcripts of the interviews were analyzed to identify themes using conventional content analysis [33]. A codebook was developed following line-by-line coding three transcripts (10% of the interviews) by two independent coders, discussed iteratively by the research team, then finalized and uploaded into NVivo. The transcripts were coded in NVivo software to apply the codebook and identify emergent themes from the data using a set coding protocol. The coders (A.A and J.L.) met regularly to discuss codes and resolve discrepancies if needed, and a third coder (E.A.S.) was brought in as necessary to discuss discrepancies between coders. After double coding three initial transcripts, the inter-related reliability (IRR) score was evaluated. A Cohen’s kappa statistic of (κ = 0.92) was found, indicating almost perfect agreement between coders, after which the coders coded the remaining transcripts independently [34]. Strategies from Lincoln and Guba’s evaluative criteria were used to increase trustworthiness of the analysis, including prolonged engagement, analyst triangulation, and creation of an audit trail [35,36]. Queries were run on codes and used to help identify emergent themes.

## 3. Results

Demographic information collected from participants (*n* = 30) can be found in Table 1. The mean age of participants was 26.8 ± 7.92 years. The majority of students identified as female (70%), identified as white (70%), and did receive financial aid (80%), yet did not have a meal plan (73%). Most participants (90%) lived off campus, while the academic status of students was more closely divided with 17 undergraduate participants (56%) and 13 graduate participants (43%).

Nine interrelated themes and fourteen subthemes emerged from the data related to the research objectives. Three themes and six subthemes are related to the first research question about the meaning and lived experience of FI. Four themes and four subthemes emerged from the data related to the second research question on facilitators and barriers to food access. Two themes and six subthemes were found related to the third research question to explore proposed solutions to food insecurity based on perceived quality and effectiveness of current resources. Each theme is described further in the following sections, supported by quotes from the college student study participants.

### 3.1. The Meaning and Lived Experience of FI (Table 2)

#### 3.1.1. Meaning of FI for Students and Perceived Definitions

The meaning of FI for students emerged as a theme with the two interrelated subthemes of FI being perceived as a lack of control, and an inevitable sacrifice made in pursuit of a long-term gain of a college degree. All students defined FI in their own words during the interview, which revealed that students associated FI with a general struggle and uncertainty about their next meal. Students described lacking control over not only the quantity of food, but also choices as to the quality and types of food that can be consumed: “*I would say food insecurity is not having the ability to eat the way that you choose to eat”* (female undergraduate student). Many students related their definitions back to a general feeling of lacking: lack of time, lack of money, lack of employment, lack of food, all of which triggers their experience with FI. One gender queer graduate student summarized the thought process and mental strain of FI: “*I think food insecurity is like inconsistency to regular, healthy meals. You know, having to actually consider what you’re buying because of budget implications. Experiencing stress about purchasing food and the cost of food if you can even buy it. You know, maybe not always eating as healthy as you would like, as often as you would like”*. Another subtheme related to sacrifice for a future goal emerged that connected the challenges students face now as a way to ensure tangible better quality of life later on. One female graduate student described, “*It’s willingness to sacrifice in the moment to get that degree, and if that means going hungry for a few days, a lot of people will be willing to make that sacrifice”*. Other students related their hardship with FI to an optimistic insurance that the future would be better for themselves and their family: “*I know that at the end of the day, when I get that job, when I get that degree and it opens doors for me to get to the place I want to go to- that all this will just be a story. And my children will not have to go through this. So I just keep being hopeful that it can only get better for now”* (female graduate student).

**Table 2 ijerph-19-12952-t002:** Emergent themes and sample quotes related to the meaning of FI and the way it impacts lived experiences and coping strategies and food decisions.

Theme	Subtheme	Sample Quote
Meaning of FI for students	Feeling of uncertainty and lack of control	“I feel like food insecurity to me is not knowing when you would truly have a meal again beyond just a bite or two if you even get that. And the struggle of figuring out if you’d have the money or the means and everything that goes into achieving getting the food. Like, that process, the struggle of doing that. If you’re struggling, I feel like that’s insecurity enough for me” (female undergraduate student)
	Short term sacrifice for long term gain	“If it came up, I would be willing to sacrifice food security to continue education, you know, for the degree at the end to make things better. So I can see a lot of students who might not be able to afford food, but are unwilling to sacrifice their academics to find better employment” (male undergraduate student)
Impact of FI on lived experiences	Compromised mental and physical health	“When I am eating the things I should to help me feel good, I can tell the difference because I slept better and felt more energized. I had better energy to do some of the workload of the research. But now I feel like I have less energy and am tired. And I feel like a big part of that is what I’m eating” (female undergraduate student)
	Feelings of isolation from limited social interactions	“I’m not as social as I could be if I had money to go out to eat with friends and to hang out that way. And even like having people over to eat or cooking for them, that also becomes like a scarcity thing, where like I’m sort of like hoarding my food because I can’t afford to host” (female graduate student)
Impact of FI on coping strategies and food decisions	Skipping meals as a matter of necessity	“I’m just trying to make my meals stretch. So, like twice a day, instead of like three times, I will skip lunch or something. Just so that I can stretch the food I have longer because I know how much I have with my budget and how long it’s going to stretch to and I need it to stretch until I get paid again. (female undergraduate student)
	Fear of running out of food influences food purchasing and preparation behaviors	“And though I’ll order like one thing, I’ll normally like eat half of it and save it the other half for the next day. So I’m even like rationing this like one meal that I’ll get from them, so that it will be actually two meals” (gender queer graduate student)

#### 3.1.2. Impact of FI on Students’ Lived Experiences

The theme of the impact of FI on students’ lived experience involved two subthemes of compromised mental and physical health and feelings of social isolation due to limited social interactions. The constraints of FI on food quality and quantity negatively affected students’ mental and physical health outcomes, reported as increased tiredness, stress, and weight gain or loss. Many students made connections among the factors of underlying mental health challenges, quality of food consumed, and impacts on academic performance. Some students described how brain function and energy levels were affected by their eating patterns, particularly in regard to improved outcomes when students were eating what they considered “healthy” foods versus less desirable outcomes when consuming “processed” foods. The mental and physical manifestations of FI also impacted students’ academic performance, specifically their ability to focus and do well in classes. Many students cited how attending lecture on an empty stomach or taking a test without a nutritious meal impacted their ability to be fully present and succeed. One student described this relationship between FI and class performance, linking food with mental and physical health: “*[Food insecurity] definitely impacted my academics in undergrad because there were times that I’d be in lecture, and I’d be like wow I’m really hungry right now, but I don’t know if I can go eat, so it definitely distracted me from the bigger thing that I was supposed to be focused on”* (male graduate student). In addition, FI was found to impact students’ lived experiences through feelings of social isolation. For many students, social gatherings around food were limited by budgetary restrictions, which was difficult considering many interpersonal interactions involve eating together. FI was shown to be an isolating experience as students had to choose between eating out and spending time with friends versus saving money and preparing their own food. One student described the experience of going out to eat with friends but sacrificing her enjoyment because of financial constraints: “*My friends wanted to go to like Red Lobster, and they all had money, but I just went and sat and got free rolls and water, because I didn’t have the money for it”* (female undergraduate student).

#### 3.1.3. Impact of FI on Coping Strategies and Food Decisions

The theme of food coping strategies and decisions to navigate time and financial constraints was characterized by two subthemes: skipping meals as a matter of necessity, and the fear of running out of food influencing food purchasing and preparation behaviors. Skipping meals was common subtheme as a matter of necessity rather than a choice, but reasons for developing this coping mechanism were variable. Most students attributed skipping meals to constraints on time or money, as one female undergraduate student reflected “*if I don’t eat, it is almost always because of my schedule”*. Many students reported consciously eating fewer meals and cutting down on portion size to spread out limited food resources and make food stretch. Another female graduate student described how financial burden associated with FI influenced her skipping meals and reducing intake*: “I’m not even eating ramen noodles. I’m skipping meals, like there’s no access. So I’m like living on loans right now, which is also a really uncomfortable place to be”*. Coping strategies related to the subtheme of fear of running out of food influencing food preparation and purchasing behaviors included rationing, hoarding, and portioning out food to manage a limited food budget. For some, meal planning and skipping meals were intertwined, as one student described*, “I won’t have enough time to go off and do grocery shopping because of this, thus and this, so then I’ll have to ration my food out even more than just having two meals a day or even one meal a day with a lot going on. That will stretch it out further until I could go grocery shopping”* (male undergraduate student). In this way, making food last longer by stretching resources was a way to cope with not just strained finances, but also time constraints. Lack of time and FI were found to be connected as students experiencing FI had less time to prepare and purchase food because of other responsibilities such as work and academics. Further, fear of running out of food impacted food purchasing such as what types of things student thought of when deciding what food to buy. Most students coped with FI by buying frozen meals, pre-prepared food, bulk food, and lower cost food. One student described their food purchasing in terms of buying cheap, shelf stable items and stocking non-perishables in order to quell the fear of running out of food: “*I did stock up on non-perishable beforehand, you know, I’ve always got rice beans and dried spaghetti in the in the cupboard. Just because there have been times in the past when there wasn’t food, so I know I would have that”* (male graduate student).

### 3.2. Facilitators and Barriers to Students’ Food Access (Table 3)

#### 3.2.1. Social Networks Facilitate Food Access

The theme of social networks as a facilitator to food access emerged from the data as the majority of students reported how other people in their social network impacted their access to food in some way. Students described how living with roommates, family, and friends could sometimes be a source of food and support: “*I used to have this friend and she would cook me dinner a lot. And it was always nice hanging out with her and like she would always cook so you know, having friends around you that you know you don’t have to pay for anything”*. The sharing of resources, groceries, and money among social networks to buy food was common among students. One student explained how her food access and financial priorities were impacted by the safety-net of her social network to access food*: “I know that I have the option of going to my boyfriend or my family members, if I need food. So I’d rather pay bills and not have late fees or people coming to my doorstep demanding money from me”*. (female undergraduate student). Both graduate and undergraduate students mentioned asking for financial and food support from family as a way to navigate not having enough money for food, but it was often not a preferred choice*: “It’s like I don’t want to ask my family members, but I also know that if I really needed to I can”* (female graduate student). Another influence from family and social networks is learned skills and strategies of navigating constrained budgets, particularly if the student came from an FI household, “*My mom always had to be really savvy with food, and so I feel like I’ve had to be pretty savvy with food because it is important to me to eat healthy and so it’s just trying to find ways to do so”* (female undergraduate student).

**Table 3 ijerph-19-12952-t003:** Emergent themes and sample quotes related to facilitators and barriers to food access.

Theme	Subtheme	Sample Quote
Social networks facilitate food access	N/A	“Because I have roommates and they buy food, then if I don’t have any food they’re like you can just eat whatever I have, because their parents give them money to go to the store and everything. And I will say I need to wait till my next paycheck and go to the store and get more food” (female undergraduate student)
Impact of on-campus resources	N/A	“The [on-campus pantry] is very encouraged, and I like that it’s now permeating the discourse on campus. They are recognizing “Hey, we have food insecure students. Here are the resources for you.” instead of just like, you know, shoving it under the rug or tucking it aside. So these issues are talked about. I think that invites me to feel more confident about using these resources on campus” (female undergraduate student)
Financial burden and priority of expenses	Food is the last priority over other student expenses	“Would I rather be hungry or homeless? Would I rather be hungry or hit my credit score X amount of points? You know, for people like me, I’m a young adult. I have my family, but we have dreams. We want to like get a house at some point, like do better and you can’t do those things if you have horrible credit or, you know, other things, financially, that are wrong. And so you have to make these really tough choices that usually end up always trickling down to affecting how you eat” (male graduate student)
	Scholarships, financial aid, and employment do not cover the basics	“I think a lot of the issue in general just tends to be that for a lot of students, college is extremely expensive, especially if you’re coming from a lower income family. Some scholarships just do not extend enough. And there’s also the issue that a lot of jobs are not available first. Like you need to have access to a car, and a lot of on campus jobs do not really pay enough. They’re like $8.25 an hour which is not very much, and you can only work 20 h a week on campus as a student. And it can be very difficult for people to make ends meet on $8.25, an hour, 20 h a week” (gender queer undergraduate student)
Stigma and social comparison	Feeling ashamed of using resources or seeking help	“I think maybe just from like a peer standpoint, they see somebody in like a random class of theirs or something and they’re like, oh, they think I’m poor now or like, I don’t have any money or my parents don’t have any money just don’t have any money to like send me for food and stuff like that, um, yeah, I mean I guess that’s probably how I felt at first” (female undergraduate student)
	Comparison to others perceived to be worse off	“I won’t utilize that just because you know, I can go, you know, a meal or two without food and be okay. But there are people out there who literally don’t know how they’re going to eat and are way worse off. So I absolutely don’t want to take away from anyone else” (female undergraduate student)

Social networks as both a resource for food and a responsibility to provide for also emerged from the data. While some students utilized their network to receive and give food, other students had a responsibility as parents and caregivers to care for and provide food for others. One student described her experience with FI as the most able-bodied adult in the household: “*It falls on me to kind of be the one to prepare food, so you know, being very careful about how I spend the money, how I budgeted and is this going to last, is this going to be sustainable”* (female graduate student).

#### 3.2.2. Impact of On-Campus Resources

The theme of food resources such as the on-campus food pantry acting as a facilitator to food access emerged from the data through providing students with a more consistent food supply. Two thirds of the students interviewed (*n* = 20) were aware of the on-campus pantry, and one third (*n* = 10) had actually used the resource. The students who knew of or used these resources had generally positive comments about how using the food pantry (which was identified as the primary on-campus food resource) enhanced their food access. Students explained how having increased availability of food through the pantry was a supplement in between grocery shopping, giving them “*a little cushion to what I already had in my pantry, and extending the resources a little longer until I did have the opportunity to have a friend take me to the grocery store and get food again”* (female undergraduate student). Particularly when access to food was limited due to academic breaks and financial constraints, the on-campus food pantry helped fill in the gap: “*I think the pantry is a huge help, it really saved me between fall and spring semester, because I had too much winter break at the end of my financial aid refund and two people to feed”* (female graduate student). Increased normalization and awareness of food resources among students was associated with decreased stigmatization of the food pantry and increased utilization. One student described that out of the various resources available on campus, she chose to use the food pantry, “*because it’s advertised and all you really have to do is fill out a quick survey, and then you walk in and made an order form and they’ll give you your order and you’re good to go”* (female undergraduate student). Also, the quality of the food offered through on-campus resources was important, as students expressed they were not in need of more pasta and cans, but rather more fruits and vegetables that they perceived as being more expensive and could not purchase themselves. One student remarked that while the pantry was helpful in ensuring a consistent quantity of food, the nutritional quality could be improved: “*I feel like it really like needs to be focused on you know, healthy fresh options. Like everyone can afford Ramen if they have to, everybody can afford pasta, like, that’s not what we’re lacking on”* (female graduate student).

#### 3.2.3. Financial Burden and Priorities Posed a Barrier to Food Access

One theme throughout the interviews was how food access was ultimately impacted by the financial burden of higher education, putting a strain on how much of the budget can be allocated to food. Subthemes included the food as a last financial priority and feeling like scholarships and financial aid do not extend enough. Other expenses such as rent, electricity, utility bills, car payments, and care for dependents or pets were priorities over food. One female undergraduate student explained how housing expenses were most important: “*it’s a big expense that has to come first, because that’s something that I can’t not pay. I can get away with not paying or reducing my grocery bill. I can’t get away with not paying my housing”*. Students described how their grocery or meal budget often came in last place or was sacrificed completely to pay for living expenses. Food was described as being “*put on the backburner”*. In describing the order of financial priorities and coping strategies of FI, one student commented “*I’ve reduced my bills to such the point where I’m just paying for the basics, you know, kind of in order to go to school. I’ve got to have Internet in order to even have any kind of employment, I have to have a phone and have to have a roof and electricity, things like that. So I’ve reduced my bills to just what is absolutely necessary to keep going. All of those would take priority over getting food”* (male undergraduate student).

A subtheme related to financial burden was how scholarships and financial aid do not extend far enough, as any funding alleviated from costs of attending college did not always cover the food budget. Many students expressed constantly having to worry about having enough money for food as a difficult daily experience. Some students described having to live on student loans to buy food, or living paycheck to paycheck, even having to go in debt to buy groceries. One female graduate student shared that she was living off the scholarships and financial aid she received, but yet her monthly stipend was “*barely enough to live on”*. Employment status was also a subtheme that directly related to FI and food access, as the amount of time students could work directly impacted their food budget, particularly for students from low-income backgrounds. Some students reflected on how their level of FI fluctuated with their employment status and how many hours they could work. One student shared how their financial situation affected all levels of food access and ultimately trickled down to consumption: “*Most of the time I go paycheck to paycheck, so there’s been times that I’ll look in my bank account and be like ‘oh this isn’t good,’ and I don’t really have a whole lot of food in my apartment. So it’s a lot of, ‘well I guess I’m having chips for dinner’ or ‘I guess I’m having pasta again for the fourth time in a row’”* (male graduate student).

#### 3.2.4. Stigma Acts as a Barrier to Accessing Food

The theme of stigma emerged as a common barrier to food access, which was characterized by the subthemes of feeling ashamed of using resources and asking for help and comparison to others perceived to be worse off. One graduate student described how the stigma of being labeled as someone needing help conflicted with their self-expectations as an independent, adult college student: “*In the sense of control, you know, a lot of us graduate students are adults that have families or are married or at least something that we feel kind of embarrassed to have to discuss those things or admit like, I haven’t eaten or I’m having trouble eating. It’s a hard conversation to have”* (male graduate student). Embarrassment, weakness, and vulnerability were also emotions related to the barrier of stigma. Another student shared the connection between FI and stigma more generally in its impact on food access: “*I think it also has to do with embarrassment. I think it has to do with kind of, you know, just the word insecurity just makes me think of like being not willing to ask for help, even though you need it”* (female graduate student).

In addition, the subtheme of comparison to others perceived to be worse off was prominent as a barrier to student use of resources. Many students described their use of food resources in the context of comparison to others and feeling like they were undeserving to those who were “*way worse off”*. One student voiced the concern that by using food resources, “*I’m taking from people or that I’m going to be looked down upon”* (male undergraduate student). Students commonly compared their FI and need for resources to community members with more visible need*: “You’ve got homeless people walking up and down on the side streets. And you see those folks and then you’re like okay, well, I know they need help because they need food. So I would rather, you know, let those folks go out there and be able to get those resources”* (female undergraduate student). Despite campus resources being an overall facilitator of food access, the underlying message was using the food resources would be taking from others who may need it more: “*I think that’s sort of a barrier for a lot of people to overcome when they have food insecurity is like asking for help is already such a leap, so it almost just feels like, you know, I need this food, but I don’t really want to like take something from somebody else”* (female graduate student).

### 3.3. Solutions for Addressing Food Insecurity (Table 4)

#### 3.3.1. Food Access Solutions

In efforts to gain insight on student-centered solutions to food insecurity, participants were asked questions on their perceptions of how campus food insecurity should be addressed. Within this, two themes and six subthemes emerged. The two themes that emerged included food access solutions and information access solutions. In reference to food access solutions, students discussed three additional strategies/subthemes that would improve their food situations including: food scholarships, financial assistance for housing and other basic needs, and increased access to the Supplemental Nutrition Assistance Program (SNAP).

**Table 4 ijerph-19-12952-t004:** Emergent themes and select quotes of students related to proposed solutions to FI and areas of improvement to increase efficacy of food resources.

Theme	Sub Theme	Sample Quote
Food access solutions	Food scholarship	“If financial aid were to give students money for groceries of some sort, since a part of education is also to make sure we have food to eat, I think it would be nice… If there was like a guaranteed stipend that college students are going to receive X amount of money for groceries” (gender queer graduate student)
	Financial assistance for housing and other basic needs	“Financial resources were huge for me, so that would be what I would feel is most helpful. Because I was homeless, at one point… So more knowledge about like social services and housing for students in distress would be huge because I might not be here having this conversation right now if I didn’t have a friend to take me in and let me finish my undergrad” (female graduate student)
	Increasing access to the Supplemental Nutrition Assistance Program (SNAP)	“I think that maybe some seminars on how to connect students with social services that they need like SNAP… things like that would be really useful. Maybe having some event or something with some sort of takeaway card or seminar to be like here are things that you can do to improve your situation and here’s how to access them” (male graduate student)
Information access solutions	Education on nutrition, budgeting, and cooking	“I would want more things online for budgeting and meal planning. Because it took a long time for me to figure out how to meal plan appropriately. And I feel like a lot of students, even if they are in hardship, find it really hard to figure that out” (female undergraduate student)
	Centralizing and educating about resources	“We need a service online to help students find scholarships and grants and things like that. If there was a service that could narrow the many resources down to the things you’re eligible for and this is what you need to do to apply for it, that kind of thing would help save time and money” (male undergraduate student)
	Destigmatizing and increasing awareness of FI	“It helped me having this kind of interview and these questions on this specific topic of food insecurity of the students on the campus, because from my standpoint, I thought that this was just my problem at first. But when I saw this kind of topic happen, I feel like it’s not only me that has this problem happen, its also like other students too, its kind of good to know many of us have this kind of situation” (female undergraduate student)

Students proposed the idea of a university-funded food scholarship to financially aid in affording food and prioritizing food security as a vital part of education. One student expressed the utility of this solution in light of the expenses and fees required to attend college by saying “*it would be nice to have a place that can help fund students being able to afford full groceries every month, or just small little funds that can get undergraduate and Grad students get the food that they need”* (male graduate student). This solution would alleviate the money spent on food both on and off campus, which students described as lifting a burden that could allow for money to stretch to other basic needs like rent. One student described how providing money for food was a responsibility and moral imperative of the institution: “*as such a huge university, I just feel like they should almost provide a free meal plan or something like that. No kid should like be dropped off at college and not know where they’re going to get their next meal to eat, you know, because they have so much other stuff going on between their social life and school and probably a job as well”* (female graduate student). Another student discussed the logistics of this solution with the detail that the food stipend could be “*maybe $10, $15 a month, something like that. So, if you are in dire straits, then you can tap into that, but if you aren’t then you wouldn’t use it and you can save it for later. You have that backup plan and it still feels somewhat self-sufficient because you’re going out and getting food, you’re not going to a pantry”* (male undergraduate student).

Beyond providing students with financial resources for academic costs such as textbooks and tuition, students expressed the need for other basic needs assistance such as housing, bills, and health expenses. As food insecurity and financial strain are intricately linked, some students expressed that the solution to addressing FI lies in a holistic approach to student support through supplementing the costs of basic needs. One student described the order of priorities and justification for basic need financial assistance: “*most of the time, food kind of takes the back burner I guess because, you know, you need housing, clothes, and water. You need gas, car payments, insurance, whatever else payments you have. You can go three weeks-ish before you die of hunger. So you can go a while without it, it’s not like the most pressing thing. So if it’s just like for a week or a few days you can kind of feel okay with not eating for a while and using the money for something else”* (male undergraduate student). Specifically, in the context of rising rent prices and the difficulty in securing housing, students described how FI could best be addressed by looking at the intersections of mental, emotional, and physical needs.

While some students were more familiar with SNAP than others, many remarked that for SNAP to be most effective, college student access needed to be improved through increased education and enrollment in the program. After discussing her positive experience with receiving a SNAP Electronic Benefit Transfer (EBT) card, one student reflected that, “*I think they should promote that EBT is an option for college students. Because there’s probably people on campus that were going through the same thing that I was that didn’t realize that it was an option”* (female undergraduate student). Education and promotion of SNAP could occur through a variety of modalities from live trainings to online workshops. Often students who had not successfully received SNAP benefits reflected frustration with the barriers to entry and accessible information about how to enroll. While a student who was able to receive SNAP felt like it “*changed my world because I didn’t have to worry about how afford food for a month when EBT was available,”* allowing them to feel empowered in their food purchasing (female undergraduate student).

#### 3.3.2. Information Access Solutions

In addition to the more tangible financial access solutions that students proposed, they also mentioned several educational strategies where having additional access to information could support their efforts to enhance their food security status. The information access solutions theme has three subthemes that provide examples of the types of information students reported would be useful, including: education on nutrition, budgeting, and cooking; centralizing and educating about FI resources; and destigmatizing and increasing awareness of FI.

Students expressed an overall desire to have increased education on a variety of topics related to food security and promoting healthier, balanced lives as college students. Many students shared their relative lack of knowledge about budgeting and developing the skill while in college was a priority to address FI: “*I definitely feel like somebody could sit down with me and tell me a little more of how to make a balanced meal on a very tight budget. That would be nice”* (female graduate student). The exact level of nutrition and cooking education varied by student, but a general interest in more seminars and education on these topics in the context of constrained food budgets was shared to empower students. Education on these topics was not discussed only in the context of a class, but also in terms of individual counseling through, “*someone you could talk to if food situations are getting hard. But it’s not a therapy appointment, because you don’t get a lot of those. It’s like, someone you can talk to that you can run by your food budget, things like that, to talk about how can you optimize this or that. Because you know you’re in dire-ish straits, but you don’t want to waste a therapy appointment on it with someone who might not really know how to do that”* (male undergraduate student).

Students also wanted information about these resources in a central, easy to find location. At this large institution, students discussed the need to centralize and educate about resources through both on-line and in-person services. In order to navigate the overwhelming and often overlapping support systems, one student expressed the desire for a specific person to act as a basic needs coordinator to help navigate food resources: “*kind of like an academic advisor, but not for classes- for food. So they’d be assigned whatever subset of the student body and there’d be a bunch of them* (male graduate student). Continuing in this theme, one student described how centralization could happen through educating the faculty and staff that have student interface to be well aware of FI and the resources to address it: “*I think that advisors and people that everybody talks to should know about how to help. Because I think a lot of people don’t know about the resources and are not educated with that knowledge of where to go and what to do with it*” (female undergraduate student). Additionally, education about resources could be funneled through specific university-sponsored events as one student described, “*I feel like it might be really helpful if the day we have orientation, we have more resources that they provide to us in terms of like where we can get food, these kinds of societies, or resources are available to get food or get help with this kind of food related subject. That might be really helpful when we first got to here and we don’t know anything about that”* (female graduate student). Thus, the effort to find and secure resources could be shifted from the student to the institution through educating the university community and streamlining accessibility.

Finally, the subtheme of destigmatizing FI and increasing awareness of the prevalence of food insecure students on campus was discussed as an institutional culture shift necessary to address the issue. Many students identified the duality of isolation and normalization of FI such that it was not widely discussed without sounding like charity or an acceptable part of the college experience. One student described that their ideal solution to destigmatize FI would include “*more, presentations, group speakers, something like that because I know there hasn’t been a lot of those…Just having more ways for people to talk about it without feeling like you know they need to keep it to themselves or only tell like a couple close friends or family members. Because, like, it shouldn’t be normalized, but it should be okay to talk about it, I guess* (male undergraduate student). Promoting events on-campus as “*free food”* instead of “*food insecurity resources”* or making historically stigmatized spaces like the food pantry into a community gathering place on campus could help with changing terminology and thought-processes behind providing equitable support.

## 4. Discussion

This study explored the meaning of FI for college students and the impact of FI on students lived experience and coping strategies and food decisions within the intersecting influences of barriers and facilitators to food access. This research also identified a variety of proposed solutions to FI in response to perceived efficacy of pre-existing resources. The findings that FI impacts many facets of a college student’s lived experience and food decisions is consistent with the literature which reports that the daily stigma and challenges of FI that are endured for an overarching desire to attend college and obtain a higher education degree [26,37]. Insights from this data revealed how students made meaning of FI as an inevitable part of paying for college, which is consistent with the literature that has identified the simultaneous normalization of FI in higher education [38].

To date, only a small number of studies have explored FI among college students using qualitative approaches, and many have focused on the physiological, psychological, and academic consequences at the individual level [24,25,26,27]. The relationship between FI and mental health, academic performance, and emotional health has been identified in qualitative and quantitative studies among college students [12,13,17]. This association was reinforced in the present research findings that students reported FI fuels a negative cycle of poor diet quality and quantity which induces poor mental and physical health outcomes.

The evidence is well established to support that education is a social determinant of health and can improve health outcomes [39]. However, for college students in this study, the linear path of advancing education and improving health outcomes is undermined by the detrimental effects of FI on mental health, physical health, and academic performance [40]. This means that while students in this study conceptualized FI as a short term sacrifice for a long term degree, it is currently unknown if longer-term psychosocial and physiological consequences of FI among current college students may have lasting effects and may perpetuate a cycle of negative outcomes [25]. Future research with more robust, longitudinal study designs is needed to identify longer term impacts of FI during formative years such as college-age.

The barriers to food access identified by college students experiencing FI have also been established and vary from campus to campus, though social stigma and financial burden are common across many colleges and universities [10,23,25,27]. One previous qualitative study on students with FI in Texas identified several themes related to barriers to food access that were consistent with the present research including the financial burden of FI and priorities that outweighed hunger concerns [27]. The contemporaneity of these themes, particularly students feeling like they are not deserving of help and that others have it worse off, demonstrates the pervasiveness of stigma and ongoing challenges for students to obtain the help they need. The barrier to accessing resources by rationalizing that others have it worse off is also reflected in the larger food insecure population [41]. The concept of time poverty, which is defined as the lack of available time to pursue discretionary activities due to compounding responsibilities [42], is evident among college students who are faced with a variety of temporal constraints related to their schedules and daily activities that severely limit the time spent on purchasing, preparing, and consuming food.

Facilitators to food access on campus that have been reported in the literature include on-campus food pantries, faculty interactions, academic support programs, and student organizations [23,28]. One additional and multi-faceted facilitator that was identified in this study is the complex nature social system of college students and how social networks can act as both a barrier and facilitator to food access. The concept of bartering for food and informal sharing of food resources that is evident in the larger FI population [41,43] is also reflected among college students experiencing FI who commonly live with roommates. Disparities in FI have been documented among students who are parents and primary caregivers [14], which necessitates further investigation on the social structure by which FI students receive and provide food. The students’ social networks, whether living with roommates, living with a partner, or living with family introduces dynamics of reciprocal, informal food sharing networks, which may help college students when others in their network have more food access. but can be detrimental when the others in the student’s social network have fewer resources and depend on the student to provide food.

A novel aspect of this study was to explore the meaning of FI by asking students to define FI in their own words. Students expressed FI as going beyond simply having enough food to eat or struggling to have consistent meals, but also considering the quality and healthfulness of the food. The USDA Household Food Security Screener Module identifies two components of food security: the amount of food that is available and the quality or variety of food that is available [30]. The respondents in this study were consistent in that they identified both of these aspects when trying to discuss the lived experience. This can inform a better understanding of how the USDA assessment of FI on the household level can better apply to the needs and experience of college students as a vulnerable population. In addition, data from this study show the importance of considering social norms and context within students’ meaning of FI, as the internalization of FI as a “normal” experience and “rite of passage” seemed to differ from how other populations conceptualize FI [44].

Understanding the lived experience of college students with FI including the barriers and facilitators to food access is critical formative work to be able to design appropriate interventions to address these issues. Strategies have emerged within the last 5–10 years to address FI among college students such as campus pantries [23]. Future research should assess the efficacy of resources on campus through evidence-based metrics, particularly considering how utilization of the on-campus food pantry was nuanced by both barriers and facilitators. While increased awareness of the pantry acted as a facilitator to food access, utilization was overall impacted by the barriers of stigma and others perceived to be worse off.

A final distinguishing factor of this study was asking students about their views on proposed solutions to FI in context of pre-existing resources. Asking college students experiencing FI to provide suggestions for addressing their food situation and that of others could be a key to effectively informing the design of policy, educating and nutrition interventions. The proposed solutions are consistent with other student-informed solutions in the literature including on-campus food assistance programs, education initiatives, and off-campus food assistance programs [45]. However, the food insecurity solutions described this study are unique to the campus context and should be considered and implemented in light of the FI prevalence and correlates at the university.

The limitations of this study are small but should be noted. First, this study was conducted during the COVID-19 pandemic, and while we did not specifically ask about additional challenges of the COVID-19 pandemic, when COVID-relates issues came up they were included in the data used for analysis. Additionally, resources on campus changed during the course of data collection, as the university-sanctioned, on-campus pantry opened in November 2020 while interviews with students began in October 2020. Many students mentioned their awareness of this resource, though the timing of the opening may have impacted their use of the pantry. Additionally, Freshman/first year students who enrolled in fall of 2020 were not included in the sample, as they had limited lived experience as college students during the study enrollment period, and thus would not be able to appropriately contribute to the study aims. Despite these limitations, this study has several strengths including the strong methodology of data analyst triangulation, which provided a check on selective perception and illuminated blind spots in interpretive analysis [35]. Obtaining saturation at 30 students allowed for a robust breadth of responses by which to understand college student FI. Additional research, both qualitative and quantitative, is needed to assess the impact of FI resources and inform evidence-based solutions for holistically supporting students. This research can inform further studies with larger, more diverse sample sizes to investigate how FI manifests among different student populations and identities, for example undergraduate versus graduate students, or students living in on- versus off-campus housing. Interventions must take a comprehensive approach to go beyond food and address basic needs insecurities, which includes insecurity related to food, shelter, water, and safety [29].

## 5. Conclusions

FI among college students cannot be conceptualized as an isolated phenomenon, but rather a deeply complex reality with intersecting layers of influence from individual level lived experiences and perceived meaning of FI, to institutional and familial facilitators to food access, to larger structural and social barriers. The findings from this study aid in the complex understanding of the lived experience of FI for college students and identifying student perspectives on how food access can be improved. While there has been a growing body of qualitative evidence to explore the detrimental effects of FI on college students, student voices must be brought into the conversation to develop solutions. Elevating student perspectives can inform decision making and intervention strategies to mitigate barriers and enhance facilitators to food access. Higher education institutions are in a critical position to address this issue of student hunger by centering student experiences, perceptions, and feedback into institutional frameworks to best meet student needs.

## Figures and Tables

**Table 1 ijerph-19-12952-t001:** Demographic characteristics of student participants (*n =* 30).

Variable	*n* (%)
Age (Mean ± SD)	26.8 ± 7.92
Race/ethnicity	
White/Caucasian	21 (70%)
Black or African American	3 (10%)
Hispanic	3 (10%)
American Indian/Alaskan Native	1 (3%)
Asian American	1 (3%)
Gender	
Female	21 (70%)
Male	7 (23%)
Other	2 (6%)
Academic status	
Undergraduate	17 (56%)
Graduate	13 (43%)
Residency status	
On-campus	3 (10%)
Off-campus	27 (90%)
Financial aid usage	
Yes	24 (80%)
No	6 (20%)
Meal plan usage	
Yes	8 (26%)
No	22 (73%)
Food security status ^1^	
Low food security	18 (60%)
Very low food security	12 (40%)
First generation status ^2^	
Yes	12 (54%)
No	10 (45%)

^1^ Participants had to score 2–6 on the USDA 6 item screener to be classified as food insecure. ^2^ Sample size for this demographic category is *n* = 22. Missing data is reflected in the first-generation status category as the question was added to the interview guide 8 interviews into data collection.

## Data Availability

The data presented in this study are available upon request from the corresponding author. The data are not publicly available due to privacy reasons.

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
