# Peer review of "Navigating Hidden Hunger: An Exploratory Analysis of the Lived Experience of Food Insecurity among College Students"

_ijerph, 2022, doi:10.3390/ijerph191912952_

Round 1
Reviewer 1 Report
The opinion of this reviewer is the present manuscript is well structured and complete in all sections. It is an interesting result.
However, there are some suggestions that I feel I can give to the authors and that would give, in my opinion, greater completeness and scientific impact to this article. These suggestions include slight modifications in the method section.
In fact, in subsection 2.2 (Data collection) it would be convenient to include more details about the design of the semi-structured interview: composition of the panel of experts, criteria for the selection of the topics to be addressed, etc. It would also be convenient to indicate whether or not a pilot study was carried out before carrying out the interviews.
Reviewer 2 Report
Very informative research that is presented clearly and thoroughly. Of course, a larger sample size would be preferred to differentiate between the UG and Grad student experience and on- versus off-campus housing. Students being asked to define FI certainly is a novel approach that could serve as a foundation for further research across demographic groups. For example, I am FI because I cannot afford to eat at "x" mid- to high-priced restaurant, or I am food insecure because I cannot afford basic staple foods.
Reviewer 3 Report
This study investigates the meaning and impact of food insecurity on college students living experiences and coping strategies through a thematic analysis of semi-structured, qualitative interviews. The emerging themes reveal the impact of FI on students’ living experiences and food decisions, and the perceived facilitators and barriers to food access for college students. The paper is very well written, methodologically sound with clear results.
some minor typos:
l 271: replace “student” with “students”
l. 331 replace “other” with “others”
l 352: delete “an”
The results of this study are mostly confirmatory of findings in related studies. An interesting new observation is the role of social networks as both a barrier and facilitator to food access. Other important findings are that FI induces a feeling of uncertainty and lack of control and is typically regarded as a short-term sacrifice for long-term gain. The study lacks a discussion with the students about their views on the quality and effectiveness of proposed or already in-place solutions. Also, respondents could have been asked to provide suggestions for addressing food insecurity among students (see e.g. Adamovic e.a., 2022).
Emilie Adamovic, Peter Newton & Veronica House (2022) Food insecurity on a college campus: Prevalence, determinants, and solutions, Journal of American College Health,70:1, 58-64, DOI: 10.1080/07448481.2020.1725019
